# Misleading Advertising of Health-Related Products in Ecuador during the COVID-19 Pandemic

**DOI:** 10.3390/diseases10040091

**Published:** 2022-10-18

**Authors:** María Belen Mena, Ivan Sisa, Enrique Teran

**Affiliations:** 1Facultad de Ciencias Médicas, Universidad Central del Ecuador, Quito, CP 170129, Ecuador; 2Colegio de Ciencias de la Salud, Universidad San Francisco de Quito USFQ, Quito, CP 170901, Ecuador

**Keywords:** direct-to-consumer advertising, drug advertising, pharmaceutical ethics, over-the-counter medications, Ecuador

## Abstract

***Background*** Media coverage of the COVID-19 pandemic increased tuning ratings during this time. The aim of this study was to identify misleading advertising of health-related products on Ecuadorian television during the COVID-19 pandemic. ***Methods*** Television channels were monitored for 111 h in the months of June and October 2020. Verbal, nonverbal and context content were analyzed from each advertising spot according to ethical standards for the promotion of products for human health ***Results*** A total of 667 spots were analyzed. Most, 90%, involved misleading advertising of health-related products. Products for gastrointestinal conditions were the most publicized (17.8%) during the period analyzed. Newscasts most often advertised products intended to improve sexual potency (22.9%) and to a lesser degree those intended to prevent and treat respiratory problems (1.8%); this relationship was reversed when compared to general programming (*p* < 0.05). ***Conclusions*** Most of the health-related products advertised on Ecuadorian television are advertised misleadingly, with news programs having the highest number of such advertisements per hour of programming.

## 1. Introduction

Media coverage of the COVID-19 pandemic is unprecedented; the media significantly increased their tuning ratings (the number of people watching a television program) during this time [1,2]. The COVID-19 pandemic also saw the parallel development of what has been called an “infodemic”, or a massive dissemination of information related to the disease through all media [3]. A substantial portion of this information concerning drugs or products with no demonstrated activity, then its use could instead increase the risk of getting infection or its progression with an unfavorable prognosis [4]; this information was often communicated in an inadequate, sensational, or distorted manner and was often supported by weak scientific evidence and may have influenced the behavior of many people in several countries, particularly frequent users of social media, with associated potential risks related to the hazardous use of these products [5].

The media is a public service that plays a key role in informing society and promoting debate [6]. While newscasts attempt to echo governmental provisions and relevant information from the local context, commercial announcements (propaganda or advertising) attempt to position in the audience a message that can influence decision-making at the individual and collective levels [2]. Journalism has a great responsibility in our society since it plays a role that no other profession does: to report regularly and create opinions on significant issues. Abdicating from this responsibility, putting economic interests or benefits before the interests of the public, is unacceptable and unethical. In the case of newscasts, during the transmission of “official news,” advertising content is also disseminated. These spots, to a greater or lesser extent, usually fund the programming. While journalistic content (informative, entertainment, etc.) may take the format that its producers consider most appropriate, it should never disguise the presence of advertising. Citizens have the right to recognize when they are presented with a journalistic message or an advertising message to make an appropriate assessment [7,8]. To mitigate these wrong messages or misinformation, for more than two decades, the World Health Organization (WHO) has promoted several initiatives to incorporate ethical criteria for drug promotion and advertising by content regulators; however, it is seen daily that the claims made about drugs and health-related products in the media are not based on scientific evidence [9,10]. In Ecuador, the consumer protection law defines “misleading” advertising as: “Any form of information or communication of a commercial nature, the content of which is totally or partially contrary to the real conditions or of acquisition of the goods and services offered or that uses texts, dialogues, sounds, images or descriptions that directly or indirectly, and even by the omission of essential product data, induces deception, error or confusion to the consumer” [11] (Appendix A). When the media hosts erroneous content, the right to be truthfully informed is compromised and the consumer is therefore misled. Although in several countries from Latin America, there are regulations for the promotion and advertising of health-related products, in practice they are not always met [12]. False and misleading advertising of drugs can lead to serious impacts on the health of the population [13]. In the case of Ecuador, all media advertising related to food, medicines and health-related products are regulated by the Agency for Regulation and Health Control (known as ARCSA in Spanish, which stands for Agencia de Regulación y Control Sanitario). Thus, by law, it is prohibited to advertise prescription drugs in the mass media [14]. With this background, the objective of this study was to identify misleading (either abusive or inappropriate) advertising of health-related products on Ecuadorian television during the COVID-19 pandemic.

## 2. Materials and Methods

A cross-sectional study was designed in which, as part of a training course in Basic Pharmacology, virtual training was provided for four hours to 31 medical students from a public university in Ecuador; this virtual training was based on the WHO guide “Understanding and Responding to Pharmaceutical Promotion” [10], as well as on the analysis of current legal regulations issued by health, communication, and consumer advocacy regulators in Ecuador [11,14,15,16]. Between June and October 2020, each student was assigned between one and four channels of open signal television, depending on the time availability, signal, and geographic access of the evaluators; they were allowed to use the “zapping” technique, which consists of switching channels with the remote control at the will of the viewer [17]. In case television signal was not available, the programming was evaluated on the channel’s internet or radio version. A total of 111 h of television were evaluated, distributed over a first period of five days in June (three months of pandemic) for an average of 18 h per day in general programming, and a second period of seven days in October (seven months of pandemic) for three hours per day exclusively in the Ecuadorian television news segment. The advertising spaces or segments (“spots”) of products that, from the evaluator’s perspective, offered to improve health directly or indirectly or prevent disease or death were considered as a unit of analysis. Simultaneously, each advertised product was investigated for whether it had a health registration or notification with a cut-off date as of 10 October 2020, on the official website of ARCSA [18,19,20].

From each advertising spot, verbal, and non-verbal messages, including images, texts and audio, were analyzed, from which the advertised indication and the parameters of compliance with good practices of promotion and advertising in mass media were extracted, according to Ecuadorian current regulations. Each advertising spot is presented as a different unit of analysis since the same product advertisement could be perceived differently by each evaluator. A scale of three parameters was built to evaluate advertising spots, based on the ethical criteria of pharmaceutical promotion and advertising and regulations issued by the Ecuadorian Health Authority [14,16]; speeches, content, and situational context where the spot was developed were analyzed. We analyzed three parameters of each advertising spot, as follows:i.Marketing authorization (granted by ARCSA);ii.Advertised indication (the veracity of information on the indication of use, associated risks in special populations, recommendations for use in healthy or chronically ill people);iii.Use of logos, names of health professionals, offers and prizes.

When an evaluator reported that an advertising spot broke at least two components of the above three parameters, the advertisement was considered misleading. To improve the intercoder reliability a checklist was used (Appendix B). Finally, the information was verified by the leading author (MBM). When there was inconsistency in the data, the evaluators were contacted, the advertising was re-analyzed, and a common agreement was reached.

### Statistical Analysis

Descriptive statistics were used to summarize baseline characteristics. Categorical variables were reported as frequencies and percentages. Additionally, the chi-squared parametric test was used to evaluate the difference between advertising offered during general versus newscast hours. A two-tailed *p*-value of <0.05 was used to consider the statistical significance. All the statistical analysis was performed using the Android version of the CDC’s Epi Info v1.4.3 software.

## 3. Results

This study identified 667 advertising spots, corresponding to 84 products, transmitted on 28 television channels distributed in the Ecuadorian cities of Quito, Cayambe, Latacunga, Ambato and Portoviejo (Table 1).

Most of the advertisements evaluated were transmitted according to a schedule considered “family time”, with a bimodal peak of concentration in the morning (06:00–08:00, 22.4%) and the afternoon (12:00–14:00, 26.6%). Newscasts concentrated seven times more on health product offers per hour of air signal compared to general programming.

It was identified that offers were related to 84 products, of which 78% (*n* = 520) had a health registration or notification issued by ARCSA, listed as follows: 39.1% drugs, 31.4% food or food supplements, 26.5% natural products, and 2.9% cosmetics.

Table 2 shows that the propaganda of products offered for gastrointestinal conditions (colitis, diarrhea, constipation, and heartburn) was the most prevalent (17.8%) during the period analyzed, and products intended to increase energy and intelligence were least offered (6.9%).

When comparing scheduled advertising, newscasts were found to advertise products intended to improve sexual potency more frequently (22.9%) and those intended to prevent and treat respiratory problems to a lesser extent (1.8%); this relationship is the opposite when compared to regular (no newscasts) television general programming (*p* < 0.005). In analyzing the content of the products, it was found that most of them suggested indiscriminate, permanent, and unnecessary use even in healthy people without mentioning restrictions of use in children or during pregnancy or lactation (Table 3).

Evaluators considered at their own judgment that 67.2% of the advertisements analyzed were misleading (news: 65% versus general programming: 69%, *p* = 0.4).

In none of the news health segments were sponsors’ sources of funding declared. In addition, 99% of the advertisements evaluated broke at least two requirements of the law, falling into the category of misleading advertising according to consumer advocacy definitions and ethical criteria for drug promotion and advertising issued by the Ecuadorian Health Authority.

## 4. Discussion

During an unprecedented pandemic, traditional media increased both the consumption of news and the positive assessment of its news coverage in terms of credibility and trust [2,21]. Newscasts are not only a segment of “information,” but also include interludes or “commercial pauses” that house advertisements that almost entirely fall into the category of misleading publicity. In Ecuador, if a citizen seeks to inform themself through the newscasts segment, they must be ready to receive up to seven times more misleading advertising than when watching any other television program, giving the impression that they end up uninformed.

Given that through advertising not only is a product offered, but values, principles, and a way to deal with this world are also transmitted, it is concerning that around half of the local channel advertisements offer products without health registration or notification, exposing people to unnecessary additional risks [22,23]. Globally, there is an active campaign to “not spread” false news or “fake news” [24], but if the media themselves promote products that offer false promises, and up to a third of them without health registration or notification, the urgency of pragmatic regulation is evident. Local regulation does exist, but it is not enforced; in fact, this would give the impression that the media, in trying to finance the news space, agrees with transmitting misleading advertising spots in the interim, which then affects citizens in terms of their right of access to truthful information; it does not seem innocent or coincidental that a health segment, interview, or report on “the importance of the immune system in COVID-19” is followed by a series of advertising spots that offer precisely to “improve immunity,” although in essence they are placebos; it is no coincidence; it is a promotional strategy that takes advantage of the pandemic; this form of rational persuasion uses deductions, inductions, rhetoric, reference, or identity to convince people of the advantages of the advertised product [13,24,25,26].

In promotional terms, exploiting peoples’ common fears usually works very well, for in essence we all fear the same things—sickness, misfortune, our own suffering and that of others, death (ours and that of our loved ones), disappointment—which newscasts casually strive to communicate all the time [13,27]. Much of the coverage is “spectacular, sensationalist, focused on particularly desperate individual cases, accompanied by background music and headlines or titles that increase the drama of the narration and particularly shocking images; with the update of data on virus transmission and deaths day by day, along with graphs of data expressing that death and desolation” [2].

It is paradoxical that while doctors and the entire health system have a mandate to adhere to an evidence-based practice, promotional strategies are used in the media to emphasize only the “benefits” of products and invite people to consume them for their healing effects, disease modifications or symptom relief, which are not necessarily scientifically proven and without considering possible adverse reactions or risks with their use.

This study pragmatically conducted an assessment that regulators should apply prior to the issuance of advertisements for health-related products, with a methodology that can be easily replicated; randomly monitoring the media, with the support of university students, could be an innovative strategy, provided regulators are involved in the identification, control, monitoring and sanctioning of misleading advertising.

### 4.1. Strengths and Limitations of the Study

First, the present study evaluated 28 television channels and their programming which allowed for a comprehensive evaluation at a country level of the health-related products advertised on Ecuadorian television. Second, we used a well-known and objective coding rubric based on the WHO’s ethical criteria for drug promotion and advertising to assess whether an advertising spot falls in the category of misleading. Third, this is, to our knowledge the first report on this topic locally. Among limitations, we should mention that we do not measure intercoder reliability; however, to assure consistency and accuracy of content analysis, the evaluators were trained using objective criteria based on WHO and local regulations [7]. Furthermore, the first author proofread the extracted information, and any inconsistency in the data was discussed and resolved via a consensus between the evaluator and the first author. Thus, we anticipate that the effect of any observer bias would be minimal.

### 4.2. Public Health Implications

Overall, direct-to-consumer (DTC) advertising is reducing, no improving, health by increasing the use of new drugs with dubious efficacy (mistreatment) or promoting the use of drugs when no treatment is needed (overtreatment) [28]. Despite the availability since 1988 of the WHO´s ethical criteria for medicinal drug promotion, the spreading of poor-quality DTC advertising is common even in high-income countries with traditionally strong pharmacovigilance and regulations to avoid drug advertisements with false or misleading information [29]. For example, in 2015 in the United States (US) the pharmaceutical industry spent about USD 5.2 billion on new TV and print campaigns [30]. Thus, Klara et al., assessing DTC advertising of prescription-only drugs in the US between January 2015 and July 2016 found that among 97 DTC advertisements, none of them described drug risks, only 26% of them presented quantitative data regarding drug efficacy, and 13% of them promoted off-label use of the medications [31].

Low- and middle-income countries (LMICs) traditionally have faced unique challenges to build appropriate/robust pharmacovigilance systems that avoid several issues including inappropriate advertising of health-related products and self-medication [32]. The found results in the present study are not surprising; however, this finding is not due to the absence of local regulation against inappropriate drug advertising but to the non-compliance and enforcement of it. Our findings should call the attention of the Ecuadorian Ministry of Health and other stakeholders to revert this situation. Some initiatives that could be implemented locally and in other LMICs in similar condition are:(i) increase well-trained personnel to support pharmacovigilance activities [32], (ii) improve education among health care professionals/students regarding drug marketing tactics [28], and (iii) enhance collaboration between different government agencies to support pharmacovigilance activities [32].

## 5. Conclusions

In Ecuador, a television viewer has more exposure to misleading advertising when watching an hour of newscasts than during any other program. The products that are most promoted seek to persuade the consumer through the health problems generated by the COVID-19 pandemic. The advertising spots analyzed have at least one element that would categorize them as misleading advertising, according to the ethical standards for the promotion of products for human health and Ecuadorian regulations; it is essential to regulate all kinds of advertising, health segments, and public reporting and to make transparent any potential conflicts of interest that finance the health segments of the newscasts. If the health authority, consumer protection agencies, market power control entities, and the like do not become involved, there will be no change. The misleading advertising of health-related products during a pandemic takes advantage of the fears of people that are mainly looking to newscast to be informed; however, instead, those programs, through the sale of publicity spots, are broadcasting misinformation, something that might considered unethical. We must learn from our current national and global failures in tackling the COVID-19 pandemic; otherwise, we will have wasted invaluable lessons learned for the next pandemic.

## Figures and Tables

**Table 1 diseases-10-00091-t001:** General characteristics of monitored communication media, according to the health registration or notification of its products.

General Characteristics	Marketing Authorization of the Products	
	No–*n* (%)	Yes–*n* (%)
Private media (*n* = 425)	95 (22.3)	330 (77.6)
Public media (*n* = 242)	54 (22.3)	188 (77.6)
Local broadcasts (*n* = 65)	29 (44.6)	36 (55.3)
National broadcasts (*n* = 602)	120 (19.9)	482 (80)
Spots in general programming (*n* = 257)	90 (35.1)	167 (64.9)
Spots in newscasts (*n* = 410)	59 (14.3)	351 (85.6)

**Table 2 diseases-10-00091-t002:** Advertising distribution, according to the health indication advertised on television, Ecuador 2020.

Advertised Health Indication	Total *n* = 677 *	Newscasts *n* = 420	General Programming *n* = 257	*p*-Value **
Colitis, diarrhea, constipation, heartburn	119 (17.8)	66 (9.8)	53 (8.0)	0.14
Immune system boosters	108 (16.2)	86 (13.0)	22 (3.2)	<0.001
Sexual complain/improvement	95 (14.4)	94 (22.9)	1 (0.4)	<0.001
Acne or baldness	74 (11.0)	43 (6.4)	31 (4.6)	0.52
Respiratory problems (flu, allergy, etc.)	68 (10.2)	12 (1.8)	56 (8.4)	<0.001
Antifungals (topical)	55 (8.2)	14 (2.1)	41 (6.1)	<0.001
Pain relieves	47 (7.0)	23 (3.4)	24 (3.6)	0.09
Energy and attention improvement	46 (6.9)	36 (5.4)	10 (1.5)	0.02
Others	65 (9.7)	46 (6.9)	19 (2.8)	0.13

* Values are shown as *n* (%). ** newcasts vs. general programming.

**Table 3 diseases-10-00091-t003:** Breach of good advertising practices of products for human health, according to broadcast segment on television in Ecuador, 2020.

Evaluation Parameters	General Programming (*n* = 863) *	Newscasts (*n* = 1480)
Uses images, names of health professionals, scientific associations, and regulatory authorities to promote the product	42 (4.9)	51 (3.4)
Advertises curative properties in chronic illnesses not proven by scientific evidence	73 (8.5)	146 (9.9)
Suggests the use of the product in healthy people	166 (19.2)	297 (20.1)
Induces in any way the use of the product in children or during pregnancy or lactation, or omits restrictions on this vulnerable group	177 (20.5)	318 (21.5)
The advertised indications lack scientific support	197 (22.8)	323 (21.8)
Induces indiscriminate, permanent, and unnecessary use of the product	208 (24.1)	345 (23.3)

* Values are shown as *n* (%).

## Data Availability

The dataset supporting the conclusions of this article is available in the following repository at https://figshare.com/s/8d709c37f998c46b2ebc (accessed on 10 August 2022).

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
