# Peer review of "Misleading Advertising of Health-Related Products in Ecuador during the COVID-19 Pandemic"

_diseases, 2022, doi:10.3390/diseases10040091_

Round 1

Reviewer 1 Report

Title: Misleading, abusive, or inappropriate advertising of medical products in Ecuador during the COVID-19 pandemic

I thank the author(s) for the opportunity to read their study. I have provided my comments below

and hope you find them useful.

1.      I would recommend that the manuscript is thoroughly proofread. Furthermore, there are instances where the author(s)s’ tone tends to get personal, for example, “Abdicating from this responsibility, putting economic interests or benefits before the interests of the public, is unacceptable and unethical.” I would suggest rephrasing “unacceptable and unethical” and use a more objective/neutral tone.  

2.      The author(s) operationalize abusive and misleading advertisements. However, what about ‘inappropriate’? How are inappropriate ads distinct from abusive and misleading ads?

3.      In the introduction, the author(s) highlight media/advertising ethics but why was COVID-19 chosen as the context of the study? There should be some mentioning of literature or rationale that connects media/advertising ethics to the pandemic.

4.      What are the hypotheses or research questions you are trying to test or explore? None are listed.

5.      Regarding the method, I have some comments:

a.       The materials and methods section should be placed before results.

b.      Why was Ecuador chosen as the country of study? Provide reasoning for its selection.

c.       When was the data collection conducted? Provide the dates (month and year).

d.      Being a content analysis, please provide the coding book in the appendix.

e.       Please report the intercoder reliability between the 31 coders.

f.        In the coding book, which items measured abusive, misleading, and inappropriate advertising? Also, please distinguish them in the results and do not compile them under one category.

6.      Overall, though interesting, I am not sure of what the main purpose is of this study and how it contributes to theory and practice.

Author Response

1. I would recommend that the manuscript is thoroughly proofread. Furthermore, there are instances where the author(s)s’ tone tends to get personal, for example, “Abdicating from this responsibility, putting economic interests or benefits before the interests of the public, is unacceptable and unethical.” I would suggest rephrasing “unacceptable and unethical” and use a more objective/neutral tone.  

We thank you to reviewer 1 for this comment and following it, professional proofreading was done.

2. The author(s) operationalize abusive and misleading advertisements. However, what about ‘inappropriate’? How are inappropriate ads distinct from abusive and misleading ads?

This was something we believed was obvious, so thank you for rising such an important point. We have chosen only to focus our study on “misleading” advertising. In lines 63-65 it was defined for this study that inappropriate means either abusive or inappropriate advertising.

3. In the introduction, the author(s) highlight media/advertising ethics but why was COVID-19 chosen as the context of the study?.

Once again, thank you to reviewer 1 for noting this missing background. The following paragraph was included (lines 30-38): “The COVID-19 pandemic also saw the parallel development of what has been called an “infodemic”, or a massive dissemination of information related to the disease through all media [3]. A substantial portion of this information concerned drugs or products that could instead favor infection, or an unfavorable prognosis were also subjects of media claims [4]. This information was often communicated in an inadequate, sensational, or distorted manner and was often supported by weak scientific evidence and may have influenced the behavior of many people in several countries, particularly frequent users of social media, with attendant potential risks related to the hazardous use of these products [5].”

4. What are the hypotheses or research questions you are trying to test or explore?

Please note that the objective of the study (lines 72-75) is mimic the research question that we were trying to address.

5. Regarding the method, I have some comments:

a. The materials and methods section should be placed before results.

Yes, it has been placed accordingly.

b. Why was Ecuador chosen as the country of study? Provide reasoning for its selection.

All authors belong to Ecuador, that was the only reason.

c. When was the data collection conducted? Provide the dates

It has been included in line 82 (Jun-Oct/2020)

d. Being a content analysis, please provide the coding book in the appendix.

We thank the reviewer for this comment. This information is available in the data repository described in lines 288-289.

e. Please report the intercoder reliability between the 31 coders.

We appreciate the reviewer´s comment. We did not assess the intercoder reliability of the coders in the present study. In response to this comment, we have added this as a limitation in the discussion section of the manuscript.

f. In the coding book, which items measured abusive, misleading, and inappropriate advertising? Also, please distinguish them in the results and do not compile them under one category.

As mentioned in point 2, we have chosen only to focus our study on “inappropriate” advertising. In lines 63-65 it was defined for this study that inappropriate means either misleading or abusive advertising.

6. Overall, though interesting, I am not sure of what the main purpose is of this study and how it contributes to theory and practice.

Thank you to reviewer 1 for this comment. In response to this comment, we have added a section entitled “Public health implications” in the discussion section of the manuscript.

Reviewer 2 Report

This study looks at the quality of promotion of health products on TV in Ecuador during different types of programming and finds significant problems with the promotion.

1.     There are some places where the grammar is awkward and the manuscript should be copy edited.

2.     The authors need to explain whether they were looking at just the promotion of pharmaceuticals or also other health products. Also, they need to say if they looked a promotion for prescribed drugs, over-the-counter drugs or both.

3.     In a few places the authors say that the pattern of which products were promoted was “reversed”. Do they mean that products for respiratory problems were the most frequently advertised ones during general programming? The authors also need to define what they mean by “general programming”.

4.     The authors need to provide more background about the promotion of pharmaceutical products in Ecuador on mass media, e.g., can prescription drugs be advertised in the mass media, are advertisements monitored and if so by whom, are there penalties for misadvertising, do advertisements need to contain information about safety, etc.?

5.     Line 32: What does "relevant information from the local context" mean?

6.     Did the authors have any grounds for believing that there would be more problems with advertising health products during the pandemic? If so that should be stated.

7.     Line 77: What is a "health registration" and "notification"? What is ARCSA?

8.     Why weren't there predefined criteria about would constitute misleading, abusive or inappropriate advertising. If only one person viewed each ad how was bias controlled for?

9.     Lines 106-108: I do not understand what the authors are trying to say in the sentence beginning "In this sense..."

10.  The Methods section should come after the Introduction and before the Results.

11.  There should be an ethics statement in the manuscript.

12.  Lines 174-175: The Ecuadorian regulations should be included as an appendix.

13.  Lines 184-185: I don't understand what "verified in an electronic form containing digital evidence of each advertising spot evaluated" mean.

14.  Line 203: The authors previously mentioned that the sponsors source of funding was not mentioned but what evidence do the authors have that there was financial COI?

Author Response

1. There are some places where the grammar is awkward and the manuscript should be copy edited.

We thank you reviewer 2 for this comment and following it, professional proofreading was done.

2. The authors need to explain whether they were looking at just the promotion of pharmaceuticals or also other health products. Also, they need to say if they looked a promotion for prescribed drugs, over-the counter drugs or both.

Thank you to reviewer 2 for this comment. To clarify, we included a paragraph in the introduction (lines 30-38).

3. In a few places the authors say that the pattern of which products were promoted was “reversed”. Do they mean that products for respiratory problems were the most frequently advertised ones during general programming? The authors also need to define what they mean by “general programming”.

Accordingly, “reversed” was changed to “the opposite” and “general programming” to “regular (no newscasts) television programming”.

4. The authors need to provide more background about the promotion of pharmaceutical products in Ecuador on mass media, e.g., can prescription drugs be advertised in the mass media, are advertisements monitored and if so by whom, are there penalties for misadvertising, do advertisements need to contain information about safety, etc.?

Thank you to reviewer 2, we added in the introduction the following paragraph “In the case of Ecuador, all media advertising related to food, medicines and health products are regulated by the Agency for Regulation and Health Control (known as ARCSA in Spanish by Agencia de Regulación y Control Sanitario), then, by law it is prohibited to advertise prescription drugs in the mass media [14].”, (lines 72-76) to contextualize the aim of the study.

Line 32: What does "relevant information from the local context" mean?

For us it means news.

5. Did the authors have any grounds for believing that there would be more problems with advertising health products during the pandemic? If so that should be stated.

Unfortunately, this has not been analyzed previously.

6. Line 77: What is a "health registration" and "notification"? What is ARCSA?

It was already changed to “granted” and ARCSA was previously defined in lines 73-76.

8. Why weren't there predefined criteria about what would constitute misleading, abusive or inappropriate advertising. If only one person viewed each ad how was bias controlled for?

This was something we believed was obvious, so thank you for rising such an important point. We have chosen only to focus our study on “inappropriate” advertising. In lines 63-65 it was defined for this study that inappropriate means either misleading or abusive advertising.

9. Lines 106-108: I do not understand what the authors are trying to say in the sentence beginning "In this sense..."

Thank you to reviewer 2 for mentioned this. We change it by the following text: “Then, newscasts are not only a segment of “information,” as also include interludes or “commercial pauses” that is housing advertisements that almost entirely fall into the category of inappropriate publicity” (lines 179-181).

  1. The Methods section should come after the Introduction and before the Results.

It has been corrected.

  1. There should be an ethics statement in the manuscript.

Please refers to line 256 (IRB statement)

  1. Lines 174-175: The Ecuadorian regulations should be included as an appendix.

Done

  1. Lines 184-185: I don't understand what "verified in an electronic form containing digital evidence of each advertising spot evaluated" mean.

Text was replaced by “Finally, the information was verified by the leading author (MBM)”.

  1. Line 203: The authors previously mentioned that the sponsors source of funding was not mentioned but what evidence do the authors have that there was financial COI?

Thank you for sharing this. We added “any potential” before COI (line 244).

Reviewer 3 Report

Dear Editor, thank you for giving me the opportunity to revise this manuscript.

It addresses potential misleading, abusive, or inappropriate advertising
of medical products on Ecuadorian television during the COVID-19 crisis.

My suggestions:

1. About the sentence (Introduction) "To mitigate these erroneous messages, for more than two decades, the World Health Organization (WHO) has promoted several initiatives to incorporate ethical criteria for drug promotion" ... please add a reference (or a link). Is it reference number 7?

2. Materials and Methods must precede Results

3. Address study limitations (e.g., random media assessment). Can you explain the differences with the pre-COVID period (if available). Please, discuss this issue.

4. Include a study flow-chart

5. Discussion must be completely revised. It should better focus on the study results. Please consider that this section is aimed at discuss results explaining and evaluating what you found, and showing how it relates to the available literature. Finally, it must make support to the conclusion.

4. Specify Funding

Author Response

  1. About the sentence (Introduction) "To mitigate these erroneous messages, for more than two decades, the World Health Organization (WHO) has promoted several initiatives to incorporate ethical criteria for drug promotion" ... please add a reference (or a link). Is it reference number 7?

Thank you to reviewer 3 for note this missing reference. It has been now link to cite #9.

  1. Materials and Methods must precede Results

Thank you, it was moved accordingly.

  1. Address study limitations (e.g., random media assessment). Can you explain the differences with the pre-COVID period (if available). Please, discuss this issue.

Thank you to reviewer 3. In lines 229-234 a paragraph about limitations was included.

  1. Include a study flow-chart

We are deeply sorry with reviewer 3 but we do not understand what means by “flow-chart” as our study had only three components: training, data collection and analysis.

  1. Discussion must be completely revised. It should better focus on the study results. Please consider that this section is aimed at discuss results explaining and evaluating what you found, and showing how it relates to the available literature. Finally, it must make support to the conclusion.

Once again, thank you to reviewer 3. Discussion has been enriched with strengths and limitations, as well as the public health implication of our findings.

  1. Specify Funding –

Please see line 254 – “This research received no external funding”

Reviewer 4 Report

This is a manuscript describing the inadequate and misleading advertising of medical products in Ecuador. It is well written, and the methodology is adequate. Although it is not an outstanding paper, it certainly deserves publication.

Author Response

Thank you reviewer 4 for your kindly comments.

Round 2

Reviewer 1 Report

I thank the author(s) for making the suggested changes. I am ok with all of the edits except that intercoder reliability was not captured, which is a major limitation and questions the validity of the findings. I would leave it to the journal editor to decide what would be the best course of action. 

Wish the author(s) the very best!

Author Response

We thank reviewer 1 for the positive feedback and based on his/her recommendation Appendix A has been added to the manuscript. It contains the codification used to capture the information. We hope it helps to solve the concern regarding the validity of the findings.

Into the methods section the following sentence was added: ". To improve the intercoder reliability a checklist was used (appendix A). "

Reviewer 2 Report

The authors have addressed my initial concerns but there are still some outstanding issues:

1.     Line 78: I do not understand what the authors mean by the phrase "that could instead favor infection or an unfavorable prognosis".

2.     Line 84: What are "official provisions"?

3.     Line 182: What "erroneous messages" are the authors referring to?

4.     Lines 197-199: What are the penalties for violating the Ecuadorian legislation and how is compliance with the legislation monitored?

5.     Line 202: The authors need to define what they mean by "medical products". They also alternate between using "health products" and "medical products". Health products would appear to be the more appropriate term and should be used consistently.

6.     Table 2: I don't understand how the percentages in Table 2 were calculated. The final right-hand column needs to indicate what was being compared; I assume it was newscasts versus general programming but that needs to be made clear.

7.     Line 578: “No” should be “not”.

8.     Line 585: Klara et al evaluated DTCA of prescription-only drugs whereas this study looked at a wide variety of health products.

 Joel Lexchin

Author Response

We thank reviewer 2 for these valuable additional comments, and following our responses:

1.     Line 78: I do not understand what the authors mean by the phrase "that could instead favor infection or an unfavorable prognosis".

This sentence was modified as follows: "A substantial portion of this information concerning drugs or products with no demonstrated activity, then its use could instead increase the risk of getting the infection or its progression with an unfavorable prognosis [4]. "

2.     Line 84: What are "official provisions"?

"Official" was replaced by governmental provisions as we referred to the guidelines provided by the sanitary authorities.

3.     Line 182: What "erroneous messages" are the authors referring to?

We changed "erroneous messages" to "wrong messages or misinformation"

4.     Lines 197-199: What are the penalties for violating the Ecuadorian legislation and how is compliance with the legislation monitored?

Although there are penalties and follow-up for its compliance, the authors considered that it was not the focus of this manuscript. For further details or info in the supplementary materials, all legal documents will be available (unfortunately in Spanish).

5. Line 202: The authors need to define what they mean by "medical products". They also alternate between using "health products" and "medical products". Health products would appear to be the more appropriate term and should be used consistently.

Thank you for raising this point. We decided to replace "medical products" and "health products" with "health related products". It was done in the whole document.

6. Table 2: I don't understand how the percentages in Table 2 were calculated. The final right-hand column needs to indicate what was being compared; I assume it was newscasts versus general programming but that needs to be made clear.

We really thank reviewer 2 for noting this critical issue. For an unknown reason, there were some mistakes at the moment to transfer numbers to the table, but now everything is amended.  Percentages in the first column belong to each parameter divided by the total.  Percentages in column two and three corresponds again to each parameter divided by the total, then column two plus column three are equal to column one. Also, the explanation of the p-value was added as a footnote in the table.

7. Line 578: “No” should be “not”.

Thank you, it is done.

8.     Line 585: Klara et al evaluated DTCA of prescription-only drugs whereas this study looked at a wide variety of health products.

Once again, thank reviewer 2, this clarification was included.

Reviewer 3 Report

I endorse it

Author Response

Thank reviewer 3 for your support